# SAC-Opt: Semantic Anchors for Iterative Correction in Optimization Modeling

**Yansen Zhang** [1]  **Qingcan Kang** [2]  **Yujie Chen** [3]  **Yufei Wang** [2]  **Xiongwei Han** [2]  **Tao Zhong** [2]  **Mingxuan Yuan** [2]
**Chen Ma** [1]

## Abstract

Large language models (LLMs) have opened new paradigms in optimization modeling by enabling the generation of executable solver code from natural language descriptions. Despite this promise, existing approaches typically remain solver-driven: they rely on single-pass forward generation and apply limited post-hoc fixes based on solver error messages, leaving undetected semantic errors that silently produce syntactically correct but logically flawed models. To address this challenge, we propose SAC-Opt, a backward-guided correction framework that grounds optimization modeling in problem semantics rather than solver feedback. At each step, SAC-Opt aligns the original semantic anchors with those reconstructed from the generated code and selectively corrects only the mismatched components, driving convergence toward a semantically faithful model. This anchor-driven correction enables fine-grained refinement of constraint and objective logic, enhancing both fidelity and robustness without requiring additional training or supervision. Empirical results on seven public datasets demonstrate that SAC-Opt improves average modeling accuracy by 7.7%, with gains of up to 21.9% on the ComplexLP dataset. These findings highlight the importance of semantic-anchored correction in LLM-based optimization workflows to ensure faithful translation from problem intent to solver-executable code.

[1]Department of Computer Science, City University of Hong Kong, Hong Kong SAR, China [2]Huawei Noah's Ark Lab, Hong Kong SAR, China [3]Huawei's Supply Chain Management Department, Shenzhen, China. Correspondence to: Qingcan Kang <kangqingcan@huawei.com>, Chen Ma <chenma@cityu.edu.hk>.

*Proceedings of the 43rd International Conference on Machine Learning*, Seoul, South Korea. PMLR 306, 2026. Copyright 2026 by the author(s).

## 1. Introduction

Optimization problems arise across domains such as logistics, healthcare, and finance, supporting tasks like planning, allocation, and portfolio optimization (Antoniou & Lu, 2007; Singh, 2012). These problems are typically formulated as mathematical programs and solved using external solvers such as Gurobi (Gurobi Optimization, LLC, 2026), CPLEX (Int, 2009), or COPT (Ge et al., 2023). However, translating real-world scenarios into solver-executable code often requires collaboration between domain experts and engineers. This process is time-consuming, hard to scale, and largely inaccessible to non-experts. This expertise barrier is reflected in a recent survey showing that 81% of Gurobi users hold advanced degrees, with nearly half specializing in operations research (Gurobi Optimization, 2023).

To lower the entry barrier and automate the modeling process, large language models (LLMs) have emerged as a promising solution for the optimization modeling task. This shift reduces reliance on manual formulation while preserving essential mathematical structure, making optimization more accessible and scalable. A recent survey categorizes progress in this area into three directions: *domain-specific LLMs*, *advanced inference frameworks*, and *benchmark datasets and evaluation* (Xiao et al., 2025). Our work builds on the inference framework line, aiming to generate solver-ready models that are not only syntactically correct but also semantically faithful to the original problem intent.

Despite the rapid progress in LLM-driven optimization modeling (Huang et al., 2024; Du et al., 2026; Xiao et al., 2025), current approaches still lack the ability to verify whether generated code faithfully reflects the problem's intended semantics. Most existing methods either rely on single-pass forward code generation based solely on the LLM's internal understanding (Wei et al., 2022; Xiao et al., 2024; AhmadiTeshnizi et al., 2024b; Deng et al., 2024), and apply limited post-hoc fixes triggered by solver errors (Shinn et al., 2023; AhmadiTeshnizi et al., 2024a), focusing on syntax or feasibility rather than semantic correctness. This leads to a critical gap: semantic errors often go undetected when the code executes without raising errors. For instance, a constraint meant to enforce an upper bound may be incorrectly implemented as a lower bound. Such mistakes result in code

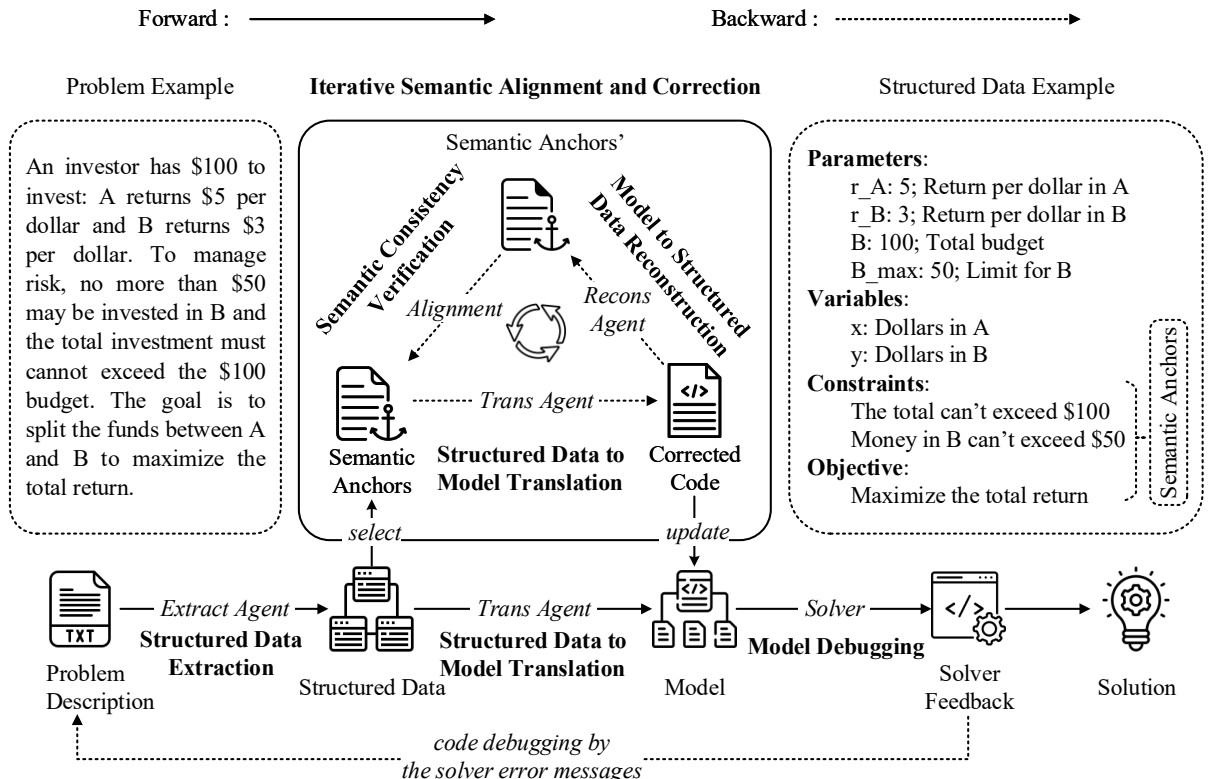

*Figure 1.* Overview of the SAC-Opt workflow. Semantic anchors represent constraints and objective in structured data. The lower pipeline shows a solver-driven workflow, while SAC-Opt uses Iterative Semantic Alignment and Correction to align code with problem semantics.

that appears functional but encodes incorrect logic, producing incorrect or misleading solutions. Since solver feedback cannot reliably signal these issues, existing pipelines are unable to detect or correct them, allowing flawed logic to silently propagate through the modeling process.

To address the limitations above, we propose SAC-Opt, a semantic anchor-driven framework for optimization modeling that performs fine-grained, iterative correction guided by problem semantics rather than solver feedback. As shown in Figure 1, SAC-Opt begins by extracting structured data from the problem description using an *extract agent*. This identifies core elements such as parameters, variables, constraints, and objective (Structured Data Extraction), which serve as the semantic foundation for later stages. We then construct an initial candidate model from the structured data (Structured Data to Model Translation), where parameters and variables are rendered using deterministic templates, and constraints and objective are generated by a *trans agent*. Unlike solver-driven approaches that equate syntactic validity with correctness, SAC-Opt establishes a backward correction loop in which semantic anchors continuously and systematically verify and refine the model, ensuring convergence toward fidelity with the original problem intent. In this work, convergence is achieved in semantic alignment, which refers to that when all anchors are correctly

represented and evidenced by the progressive decrease in semantically misaligned anchors across iterations.

The core mechanism of SAC-Opt is iterative semantic alignment, a convergence-driven process that progressively eliminates mismatches between the generated model and the original task description (Iterative Semantic Alignment and Correction). After we identify semantic anchors from structured data, typically constraint and objective expressions. For each anchor, we reconstruct its semantics from the generated code (Model to Structured Data Reconstruction) and compare it with the original anchor (Semantic Consistency Verification), using LLM-based or similarity-based checks in our implementation. Alignment is evaluated at the anchor level: each semantic anchor serves as a reference representing the problem intent. A mismatch indicates that the code does not faithfully capture the intended semantics, in which case SAC-Opt updates only the misaligned component. This anchor-driven refinement continues until full anchor consistency or a predefined iteration limit, enabling fine-grained correction without regenerating the entire model. After alignment, all components are assembled into a complete program and passed to a solver (Model Debugging). Code debugging is then applied with solver feedback, modifying the code only when execution errors occur. Finally, the corrected program is executed for the solution. In experi-

ments on seven public datasets, SAC-Opt boosts average modeling accuracy by 7.7%, highlighting the effectiveness of semantics-anchored backward correction.

**Contributions.** (1) We introduce SAC-Opt, the first optimization modeling framework that performs proactive semantic verification to detect silent semantic errors that solver-driven checks cannot capture. (2) We propose a backward, semantic anchor-guided correction mechanism that progressively aligns models with problem intent, achieving convergence through fine-grained refinement. (3) We evaluate SAC-Opt on seven public datasets and show that it improves modeling accuracy by 7.7% on average, with a 21.9% gain on the challenging ComplexLP dataset.

## 2. Related Work

### 2.1. LLMs for Optimization

LLMs show great promise for optimization, offering innovative approaches to optimize and automate modeling processes (Xiao et al., 2025; Huang et al., 2024; Du et al., 2026). A recent survey Xiao et al. (2025) categorizes this line of research into *domain-specific LLMs* (Huang et al., 2025a; Jiang et al., 2025; Li et al., 2025; Ethayarajh et al., 2024), *advanced inference frameworks* (Deng et al., 2024; Xiao et al., 2024; Li et al., 2023; Zhang et al., 2025a; Astorga et al., 2025; AhmadiTeshnizi et al., 2024a;b; Ju et al., 2024; Zhang et al., 2024a), and *benchmark datasets and evaluation* (Ramamonjison et al., 2023; Huang et al., 2025a;b; Xing et al., 2024; Yang et al., 2025). Our work builds on inference frameworks, which aim to generate solver-ready models from natural language problem descriptions. However, most existing methods often ignore and cannot verify whether the generated code reflects the intended semantics. We address this limitation by introducing an iterative correction framework that reconstructs problem intent and ensures semantic alignment.

### 2.2. Correction in Optimization

Correction in LLMs refers to the ability of a model to revise or improve its own outputs based on internal or external feedback (Pan et al., 2024; Wang et al., 2024). This mechanism has attracted increasing interest as a way to enhance reasoning accuracy and robustness without additional supervision (Kamoi et al., 2024; Zhang et al., 2025b; 2024b). Prior works AhmadiTeshnizi et al. (2024a); Deng et al. (2024); Tsouros et al. (2023) have explored using human experts or LLM feedback to refine extracted elements such as parameters, variables, and constraints, and code debugging by the solver error messages. However, these efforts focus on extraction or post-hoc debugging and overlook semantic alignment. In contrast, our method integrates semantic-level correction, ensuring fidelity to problem intent.

## 3. Methodology

### 3.1. Problem Formulation

Optimization modeling is the process of transforming a problem description in natural language $P$ into a mathematical program $\mathcal{M}$ that can be executed by an optimization solver. In its most general form, $\mathcal{M}$ comprises a decision vector $x \in \mathbb{R}^n$, a scalar objective function $f(x; \theta)$ to be minimized or maximized, and a feasible region $X(\theta)$ specified by equality and inequality constraints. For example, an optimization problem can be written mathematically as,

$$
\begin{aligned}
\min_{x \in X(\theta)} \quad & f(x; \theta) \\
\text{s.t.} \quad & g_i(x; \theta) = 0, \quad i = 1, \ldots, m, \\
& h_j(x; \theta) \leq 0, \quad j = 1, \ldots, p,
\end{aligned}
\tag{1}
$$

where $\theta$ aggregates all problem-specific parameters (such as coefficients, bounds, etc.), $g_i$ denotes the set of equality constraints, and $h_j$ denotes the set of inequality constraints.

### 3.2. Structured Data Extraction

To bridge the gap between free-form descriptions and formal programs, and inspired by the works (AhmadiTeshnizi et al., 2024b;a; Jiang et al., 2025), we first convert the natural language problem description $P$ into structured data,

$$
S = (\mathcal{P}, \mathcal{V}, \mathcal{C}, \mathcal{O}),
\tag{2}
$$

where each component corresponds exactly to the four elements of the mathematical formulation in Eq. 1. Here $\mathcal{P}$ is the set of named parameters, $\mathcal{V}$ is the set of decision variables, $\mathcal{C}$ is the collection of semantic constraints (both equality and inequality), and $\mathcal{O}$ is the objective description.

Specifically, we use an *extract agent* to extract the structured data from the problem description $P$:

$$
S = f_{\text{agent}}^{\text{extract}}(P),
\tag{3}
$$

where $f_{\text{agent}}^{\text{extract}}$ outputs a structured representation of parameters, variables, constraints, and objective in JSON format. An example of the extracted structured data is provided in Appendix A.1.

This structured data representation makes all components explicit and machine-readable, reducing ambiguity in downstream tasks. By separating parameters, variables, constraints, and objective, it enables consistency checks and modular validation. Most importantly, it supports fine-grained correction by isolating semantic elements such as individual constraints or objective, allowing errors to be detected and corrected precisely without reprocessing the entire problem description.

### 3.3. Structured Data to Model Translation

In a standard optimization workflow, the final goal is to generate executable code that can be directly run on external solvers. This code serves as the final representation of the optimization problem and must accurately capture the semantics of the original task description. Given the structured data $S = (\mathcal{P}, \mathcal{V}, \mathcal{C}, \mathcal{O})$, which encapsulates all necessary modeling elements, our goal is to convert $S$ into code $\mathcal{M}$ that preserves logical correctness and is executable without further human intervention.

Formally, we denote the overall translation process within our proposed framework as an input-output map from a structured data input $S$ to executable solver code $\mathcal{M}$. The notation specifies module interfaces and does not assume deterministic behavior for agent-based calls. To ensure modularity and reflect the semantic decomposition of $S$, we explicitly separate the output into two parts:

$$\mathcal{M} = \mathcal{M}_{\text{simp}} + \mathcal{M}_{\text{sem}}, \tag{4}$$

where $\mathcal{M}_{\text{simp}}$ and $\mathcal{M}_{\text{sem}}$ correspond to the code fragments generated from the simple and semantically rich components of $S$, respectively. Specifically, we define the code generation process as follows,

$$\mathcal{M}_{\text{simp}} = f_{\text{det}}^{\text{trans}}(S_{\text{simp}}), \tag{5}$$

$$\mathcal{M}_{\text{sem}} = f_{\text{agent}}^{\text{trans}}(S_{\text{sem}}), \tag{6}$$

where $S_{\text{simp}} = \{\mathcal{P}, \mathcal{V}\}$ includes parameters and variables that are fully specified and can be deterministically rendered, while $S_{\text{sem}} = \{\mathcal{C}, \mathcal{O}\}$ contains constraints and objective, which represent the key logic of the optimization task. These elements in $S_{\text{sem}}$ are essential, as they directly impact the correctness of the model and require careful modeling to preserve the intended meaning.

The deterministic function $f_{\text{det}}^{\text{trans}}$ uses pre-defined code templates with fixed rendering rules. For example, a parameter named `RollWidth` is rendered as: `RollWidth = data["RollWidth"]`. This approach provides consistent rendering for syntactically well-defined elements. In contrast, the semantic translation function $f_{\text{agent}}^{\text{trans}}$ employs a *trans agent* to generate code directly from natural-language sentences. This process avoids intermediate representations such as LaTeX or pseudo code, thereby reducing cumulative translation errors (Astorga et al., 2025) and simplifying downstream integration.

This separation keeps deterministic fields fixed while exposing constraints and objective as editable units for reconstruction and correction.

### 3.4. Model to Structured Data Reconstruction

Existing LLM-based optimization workflows (Xiao et al., 2024; AhmadiTeshnizi et al., 2024b; Deng et al., 2024) end once executable code is generated, and rely on solver error messages for post-hoc checks (Shinn et al., 2023; AhmadiTeshnizi et al., 2024a). However, such forward pipelines cannot detect semantic errors in the constraint or objective logic. Solvers validate syntax and feasibility but cannot determine whether the encoded logic reflects the original task intent. This limitation leads to models that may run without error yet fail to solve the intended problem.

To address this challenge, we introduce a semantic-anchored backward correction framework that leverages the extracted semantic anchors $S_{\text{sem}} = \{\mathcal{C}, \mathcal{O}\}$ to assess whether the generated code correctly reflects the original modeling intent. After producing the solver-executable code $\mathcal{M}_{\text{sem}}$, we apply a reconstruction step to recover the code's logic corresponding to the semantic anchors:

$$\widehat{S}_{\text{sem}} = f_{\text{agent}}^{\text{recons}}(\mathcal{M}_{\text{sem}}), \tag{7}$$

where $f_{\text{agent}}^{\text{recons}}$ is a *recons agent* that generates the corresponding constraints or objective anchors from the code and formats them into the same structured form as the original semantic anchors for comparison and analysis. The exact prompt design is detailed in Appendix A.2.

Because $\widehat{S}_{\text{sem}}$ uses the same anchor format as $S_{\text{sem}}$, each recovered constraint or objective can be compared directly with its original counterpart in the correction step.

### 3.5. Iterative Semantic Alignment and Correction

Based on the reconstructed semantic anchors $\widehat{S}_{\text{sem}} = \{\widehat{s}_i \mid \widehat{s}_i \in \widehat{\mathcal{C}} \cup \widehat{\mathcal{O}}\}$ derived from the generated code $\mathcal{M}_{\text{sem}}$, we introduce an iterative backward correction process to align the model with the original semantic anchors. This step constitutes the core of our iterative correction framework.

Specifically, the goal is to ensure that each reconstructed semantic component $\widehat{s}_i$ is consistent with its original counterpart $s_i \in S_{\text{sem}} = \mathcal{C} \cup \mathcal{O}$. To formalize the semantic consistency checking, we define a binary consistency verification function as follows,

$$\delta(s_i, \widehat{s}_i) = \begin{cases} 1 & \text{if } s_i \equiv \widehat{s}_i, \\ 0 & \text{otherwise}, \end{cases} \tag{8}$$

where $\equiv$ denotes semantic equivalence. In this work, we provide two alternative strategies to implement this equivalence function in our framework:

**LLM-based Verification:** (Gu et al., 2026; Li et al., 2024; Schroeder & Wood-Doughty, 2024)

$$\delta_{\text{LLM}}(s_i, \widehat{s}_i) = \mathbf{1}\left[f_{\text{agent}}^{\text{verif}}(s_i, \widehat{s}_i) = \texttt{True}\right], \tag{9}$$

where $f_{\text{agent}}^{\text{verif}}$ is a binary classifier implemented via a *verif agent* that determines whether $s_i$ and $\widehat{s}_i$ are semantically

equivalent. The exact prompt design is detailed in Appendix A.3.

**Similarity-based Verification:** (Chowdhury, 2010; Chandrasekaran & Mago, 2021; Strang, 2022)

$$\delta_{\text{sim}}(s_i, \widehat{s}_i) = \mathbf{1}\left[\cos\left(\phi(s_i), \phi(\widehat{s}_i)\right) \geq \tau\right], \quad (10)$$

where $\phi(\cdot)$ is a pretrained sentence encoder and $\tau \in [0, 1]$ is a user-defined scalar similarity threshold, and $cos$ is the standard cosine similarity function.

These two variants are alternative automatic implementations of the same consistency function $\delta$; they are not fused scores, and neither verifier is treated as a perfect oracle.

To verify the semantic fidelity of the generated model, we apply the consistency verification function $\delta(s_i, \widehat{s}_i^{(t)})$ to each semantic anchor $s_i \in S_{\text{sem}}$, comparing it with its reconstructed counterpart $\widehat{s}_i^{(t)}$. This identifies elements where the generated code fails to capture the original modeling intent. At each iteration $t$, we define the error set as:

$$\mathcal{E}^{(t)} = \{\, s_i \in S_{\text{sem}} \mid \delta(s_i, \widehat{s}_i^{(t)}) = 0 \,\}. \quad (11)$$

This error set drives the core correction loop. If $\mathcal{E}^{(t)} = \emptyset$, all semantic anchors are consistent, and we return the final model $\mathcal{M} = \mathcal{M}_{\text{simp}} + \mathcal{M}_{\text{sem}}^{(t)}$. Otherwise, we enter the correction phase, where each inconsistent anchor $s_i \in \mathcal{E}^{(t)}$ is used to regenerate the corresponding code segment:

$$\mathcal{M}_{\text{sem}}^{(t+1)}[s_i] \leftarrow f_{\text{agent}}^{\text{trans}}(s_i). \quad (12)$$

After regeneration, we apply the reconstruction function again to obtain the updated semantic anchors $\widehat{S}_{\text{sem}}^{(t+1)}$, and repeat the consistency check. This loop continues until the error set is empty or the maximum number of iterations $T_{\text{max}}$ is reached. A more detailed discussion of the convergence is provided in the Appendix A.4. Upon termination, we return the final executable model $\mathcal{M}$, which combines the deterministic components $\mathcal{M}_{\text{simp}}$ with the latest semantically aligned code $\mathcal{M}_{\text{sem}}^{(t)}$. The complete procedure of SAC-Opt is summarized in Algorithm 1.

### 3.6. Model Debugging

After semantic correction, we assemble the final model by integrating the corrected components with standard initialization and solver statements, following the setup from prior work (AhmadiTeshnizi et al., 2024b;a). The complete code is then executed. If the solver runs successfully, the optimal solution is returned. Otherwise, we use the error messages by the solver to identify and fix the errors in generated code with the original problem description as previous works (Shinn et al., 2023; AhmadiTeshnizi et al., 2024a). This process repeats until the model runs correctly or a pre-defined iteration limit is reached.

---

**Algorithm 1** SAC-Opt: Iterative Correction with Semantic Anchors

**Input:** Problem description $P$, max iterations $T_{\text{max}}$, similarity threshold $\tau$
**Output:** Corrected model $\mathcal{M}$
**Structured data extraction**
$S = (\mathcal{P}, \mathcal{V}, \mathcal{C}, \mathcal{O}) \leftarrow f_{\text{agent}}^{\text{extract}}(P)$
$S_{\text{simp}} \leftarrow \{\mathcal{P}, \mathcal{V}\}, \quad S_{\text{sem}} \leftarrow \{\mathcal{C}, \mathcal{O}\}$
**Initial code generation ($t = 0$)**
$\mathcal{M}_{\text{simp}} \leftarrow f_{\text{det}}^{\text{trans}}(S_{\text{simp}})$
**for each** $s_i \in S_{\text{sem}}$ **do**
  $\mathcal{M}_{\text{sem}}^{(0)}[s_i] \leftarrow f_{\text{agent}}^{\text{trans}}(s_i)$
**end for**
**Iterative correction loop**
**for** $t = 1$ **to** $T_{\text{max}}$ **do**
  $\widehat{S}_{\text{sem}}^{(t)} \leftarrow f_{\text{agent}}^{\text{recons}}(\mathcal{M}_{\text{sem}}^{(t-1)})$
  $\mathcal{E}^{(t)} \leftarrow \{\, s_i \in S_{\text{sem}} \mid \delta(s_i, \widehat{s}_i^{(t)}) = 0 \,\}$
  **if** $\mathcal{E}^{(t)} = \emptyset$ **then**
    **break**
  **end if**
  **for each** $s_i \in \mathcal{E}^{(t)}$ **do**
    $\mathcal{M}_{\text{sem}}^{(t)}[s_i] \leftarrow f_{\text{agent}}^{\text{trans}}(s_i)$
  **end for**
**end for**
$\mathcal{M} \leftarrow \mathcal{M}_{\text{simp}} + \mathcal{M}_{\text{sem}}^{(t)}$
**return** $\mathcal{M}$

---

## 4. Experiments

### 4.1. Dataset

To assess performance across diverse scenarios, we evaluate all methods on a suite of publicly available optimization modeling datasets, including **NL4OPT** (Ramamonjison et al., 2023), **IndustryOR** (Huang et al., 2025a), **EasyLP** and **ComplexLP** (Huang et al., 2025b), **NLP4LP** (AhmadiTeshnizi et al., 2024a), **ReSocratic** (Yang et al., 2025), and **ComplexOR** (Xiao et al., 2024). While widely used, these datasets contain substantial annotation noise, as shown in a recent survey (Xiao et al., 2025), raising concerns about reliability. To ensure consistency and fairness, we directly adopt the cleaned and standardized versions provided by the survey (Xiao et al., 2025) for all methods. These datasets span a diverse range of optimization tasks, including simple and complex problems, concrete and abstract modeling, and long-form natural language descriptions. Detailed dataset statistics are provided in Appendix A.5.

### 4.2. Baselines

We evaluate our method against a set of representative baselines covering both standard prompting and recent state-of-the-art approaches. **Standard** refers to direct single-step prompting without intermediate reasoning. **Chain-of-**

**Thought (CoT)** (Wei et al., 2022) elicits step-by-step reasoning in natural language. **Chain-of-Experts (CoE)** (Xiao et al., 2024) is a multi-agent framework where each agent specializes in a role with domain-specific knowledge. **CAFA** (Deng et al., 2024) translates problem descriptions into solver-executable code via a single-step formalization process. **Reflexion** (Shinn et al., 2023) introduces feedback-based refinement after initial code generation. **OptiMUS-0.2** (AhmadiTeshnizi et al., 2024b) uses a modular architecture to handle long and complex problems. **OptiMUS-0.3** (AhmadiTeshnizi et al., 2024a) augments extraction with correction mechanisms during parameter, variables, constraints, and objective identification.

### 4.3. Experimental Setup

To ensure a rigorous and fair comparison across all fully open-source optimization modeling baselines, we adopt a unified evaluation protocol. In all our experiments, the program uses Python as the programming language and Gurobi as the solver. Following prior work (Xiao et al., 2025), we use GPT-4o (Achiam et al., 2023) as the backbone model for all methods, and we directly report the results for Standard, CoT, CoE, and CAFA from (Xiao et al., 2025) to ensure consistency and comparability. For Reflexion, OptiMUS-0.2, and OptiMUS-0.3, we run the official open-source implementations using default hyperparameters. To control for variations in data preprocessing, all methods operate on structured data produced by a shared pipeline, more discussion about the extraction is provided in Appendix A.6. Additionally, to ensure fairness, the number of debugging attempts is uniformly set to 3 where applicable. For our method, the maximum number of correction iterations $T_{\max}$ is set as 5. The semantic similarity function $\phi(\cdot)$ is implemented using a pretrained SentenceTransformer model (all-MiniLM-L6-v2), with a similarity threshold $\tau$ set to 0.75. The source code is available at https://github.com/Forrest-Stone/SAC-Opt.

We evaluate performance based on the accuracy metric, consistent with the evaluation settings used in Xiao et al. (2024; 2025); AhmadiTeshnizi et al. (2024a;b). A solution to one problem is considered correct if the generated code executes successfully, produces the correct optimal objective value, and returns the correct optimal solution. The ground-truth values are provided by the dataset. All results are averaged over five independent runs to ensure statistical reliability and reduce evaluation variance.

### 4.4. Overall Performance

Table 1 summarizes the comparative performance of various methods evaluated under a unified protocol. Unless otherwise specified, we report the results based on LLM-based verification to measure the semantic consistency checking

$\delta(s_i, \widehat{s_i})$. A detailed comparison between LLM-based and similarity-based verification will be provided in Sec. 4.7.

Several key observations can be drawn from the Table 1. First, SAC-Opt consistently achieves the best performance across all datasets, with especially large gains on hard datasets such as IndustryOR, ComplexLP, and ReSocratic, including a 21.9% improvement on ComplexLP. Second, compared to Reflexion and OptiMUS-0.3, SAC-Opt's iterative correction introduces targeted semantic anchors alignment, outperforming syntax-level strategies and demonstrating the value of semantic-anchored optimization feedback. Third, while CoE and OptiMUS-0.2 perform well on simpler datasets, their performance degrades sharply on more complex ones, indicating that limited reasoning depth and weak feedback mechanisms fail to generalize. Finally, CoT does not consistently improve performance over standard prompting and occasionally leads to a noticeable drop in EasyLP, while CAFA yields similar results, suggesting we should design the prompt carefully.

### 4.5. Ablation Study

To better understand the contributions of individual components in SAC-Opt, we conduct an ablation study summarized in Table 2. Specifically, *w/o correction* removes the semantic anchor-guided iterative correction mechanism (Sec. 3.5), while *w/o debugging* disables the final code-level correction based on solver feedback (Sec. 3.6). The results show that removing semantic correction leads to a substantial drop in modeling accuracy across all datasets, underscoring the effectiveness of explicitly incorporating semantic anchor correction into the modeling process. This confirms their key role in aligning generated models with the intended problem semantics. Although disabling code-level debugging also reduces performance, the impact is notably smaller, indicating that while post-generation fixes can help, semantic-anchored correction is the primary driver of modeling quality. Additional analysis in Appendix A.7 further supports this conclusion by showing that improvements are more sensitive to semantic correction than to the number of code-level fixes.

### 4.6. Generalization Evaluation

Although the performance naturally depends on the reasoning ability of the underlying LLM, our goal here is to examine whether the benefits of SAC-Opt come from the semantic correction mechanism itself rather than from the strength of any particular model. The pipeline of SAC-Opt, consisting of semantic anchor extraction, semantic verification, and correction, can be instantiated with different LLMs as long as they have adequate reasoning capacity. To assess this, we further evaluated SAC-Opt using the open-source Qwen2.5-72B-Instruct model while keeping

*Table 1.* Accuracy comparisons of different methods. Methods marked with * are results directly referenced from (Xiao et al., 2025), conducted under the same experimental setting. For each dataset, the best result is shown in **bold**, and the second-best is underlined. The Impr. represents the percentage improvement relative to the second-best method.

| Method | NL4OPT | IndustryOR | EasyLP | ComplexLP | NLP4LP | ReSocratic | ComplexOR |
|---|---|---|---|---|---|---|---|
| Standard* | 61.2% | 38.1% | 70.3% | 57.7% | 73.6% | 48.4% | 42.9% |
| CoT* | 62.2% | 40.5% | 49.5% | 42.3% | 74.7% | 43.6% | 39.2% |
| CoE* | 66.7% | 31.2% | 94.4% | 50.6% | 87.4% | 71.2% | 57.1% |
| CAFA* | 68.1% | 41.1% | 71.2% | 44.5% | 50.0% | 40.1% | 46.4% |
| Reflexion | 68.2% | 49.0% | 85.8% | 43.2% | 82.4% | 76.1% | 42.2% |
| OptiMUS-0.2 | 69.2% | 43.8% | 89.2% | 45.8% | 86.5% | 75.8% | 48.9% |
| OptiMUS-0.3 | 79.8% | 54.3% | 92.4% | 52.1% | 89.8% | 81.0% | 52.2% |
| SAC-Opt | **86.8%** | **63.8%** | **96.5%** | **79.6%** | **94.0%** | **88.7%** | **58.9%** |
| Impr. | 7.0% ↑ | 9.5% ↑ | 2.1% ↑ | 21.9% ↑ | 4.2% ↑ | 7.7% ↑ | 1.8% ↑ |

*Table 2.* Ablation study of SAC-Opt. For each dataset, the best result is shown in **bold**.

| Method | NL4OPT | IndustryOR | EasyLP | ComplexLP | NLP4LP | ReSocratic | ComplexOR |
|---|---|---|---|---|---|---|---|
| SAC-Opt | **86.8%** | **63.8%** | **96.5%** | **79.6%** | **94.0%** | **88.7%** | **58.9%** |
| w/o correction | 82.9% | 50.5% | 86.6% | 63.8% | 90.1% | 80.2% | 54.4% |
| w/o debugging | 84.6% | 60.5% | 92.4% | 72.3% | 92.8% | 84.5% | 56.7% |

the extraction step fixed for fairness. As shown in Table 3, although Qwen2.5-72B-Instruct yields lower base accuracy than GPT-4o, SAC-Opt consistently provides clear improvements across all datasets. This indicates that the gains stem from the semantic correction process rather than dependence on a specific LLM, and that SAC-Opt remains effective even when paired with a less capable model.

### 4.7. Semantic Verification Comparison

To evaluate the impact of different semantic alignment strategies in SAC-Opt, we compare two variants introduced in Sec 3.5: LLM-based verification (LLM) and similarity-based verification (Sim) to compute $\delta(s_i, \widehat{s_i})$. As shown in Table 4, we report results across four dimensions: accuracy, average run time, and correction and debugging attempts. The LLM-based variant consistently outperforms the similarity-based counterpart across all metrics except debugging. It achieves higher accuracy, shorter run time, and fewer correction iterations, highlighting superior efficiency in aligning outputs with task semantics. Debugging numbers remain comparable, suggesting both methods reach a similar threshold for code-level convergence once semantic correction stabilizes.

Across the seven dataset-level accuracy pairs in Table 4, the two automatic variants are strongly correlated (Pearson = 0.962, $R^2$ = 0.925, Spearman = 0.929), and the LLM-based verifier improves accuracy by 6.94 points on average. This result provides dataset-level agreement evidence between two automatic verifier choices, but it is not a human-calibrated anchor-level validation.

Our similarity-based verification relies on a widely adopted pretrained SentenceTransformer model, selected for its low computational overhead and ease of deployment. This encoder is efficient enough to run without GPU support, allowing our method to operate on machines with limited resources. Interestingly, despite its relative simplicity, the similarity-based variant still outperforms most baselines in Table 1 on several challenging datasets, including ComplexLP and ReSocratic. This highlights the robustness of our iterative correction architecture, even when paired with lower-fidelity semantic signals. At the same time, the increased run time and correction iterations suggest that coarse similarity signals may introduce noise or misalignment, motivating future work on more accurate alignment strategies that maintain computational efficiency.

### 4.8. Case Study

To demonstrate how SAC-Opt performs iterative semantic correction, we present an example from problem *cutting stock* in the ComplexOR dataset. We focus on a constraint anchor: *"Each pattern j should generate rolls with widths that fit within the RollWidth"*. As shown in Figure 4.8, SAC-Opt begins by generating an initial code snippet for this anchor, with the error flag initialized to "". It then produces a new natural language description of the code's semantics as the reconstructed anchor and compares it against

*Table 3.* Performance comparison with and without SAC-Opt correction across different LLM models.

| Model | Method | NL4OPT | IndustryOR | EasyLP | ComplexLP | NLP4LP | ReSocratic | ComplexOR |
|---|---|---|---|---|---|---|---|---|
| GPT-4o | w/o correction | 82.9% | 50.5% | 86.6% | 63.8% | 90.1% | 80.2% | 54.4% |
| | correction | 86.8% | 63.8% | 96.5% | 79.6% | 94.0% | 88.7% | 58.9% |
| Qwen2.5-72B-Instruct | w/o correction | 80.2% | 39.5% | 77.4% | 57.5% | 89.7% | 76.7% | 40.0% |
| | correction | 85.1% | 45.7% | 84.4% | 62.9% | 93.0% | 85.9% | 43.3% |

*Table 4.* Comparison of different verification strategies. For each dataset, we report results from LLM-based (LLM) and similarity-based (Sim) methods across accuracy, run time (in seconds), and the number of corrections and debugging attempts (mean $\pm$ standard deviation).

| Dataset | Accuracy (%) | | Run Time (s) | | # Corrections | | # Debugging Attempts | |
|---|---|---|---|---|---|---|---|---|
| | LLM | Sim | LLM | Sim | LLM | Sim | LLM | Sim |
| NL4OPT | 86.8 | 83.1 | 78.43 | 156.83 | $1.13 \pm 1.70$ | $4.63 \pm 1.23$ | $0.04 \pm 0.26$ | $0.05 \pm 0.30$ |
| IndustryOR | 63.8 | 52.9 | 80.20 | 209.68 | $1.55 \pm 2.15$ | $3.72 \pm 2.11$ | $0.31 \pm 0.66$ | $0.16 \pm 0.37$ |
| EasyLP | 96.5 | 89.8 | 92.88 | 172.87 | $2.09 \pm 1.97$ | $2.18 \pm 1.95$ | $0.03 \pm 0.21$ | $0.03 \pm 0.21$ |
| ComplexLP | 79.6 | 65.3 | 40.96 | 173.76 | $1.05 \pm 1.66$ | $3.58 \pm 2.13$ | $0.12 \pm 0.41$ | $0.04 \pm 0.26$ |
| NLP4LP | 94.0 | 89.6 | 73.97 | 208.67 | $1.17 \pm 1.70$ | $4.49 \pm 1.50$ | $0.03 \pm 0.24$ | $0.05 \pm 0.30$ |
| ReSocratic | 88.7 | 82.2 | 79.85 | 152.98 | $1.18 \pm 1.81$ | $4.22 \pm 1.80$ | $0.05 \pm 0.27$ | $0.09 \pm 0.40$ |
| ComplexOR | 58.9 | 56.8 | 42.02 | 66.58 | $0.73 \pm 1.68$ | $2.36 \pm 2.29$ | $0.27 \pm 0.47$ | $0.18 \pm 0.40$ |

the original anchor. In this example, the generated description correctly summarizes the faulty code implementation logic but fails to capture the original intent of the anchor, prompting the error flag to update to *Yes*. SAC-Opt then enters its anchor-guided correction loop, where new code is generated, reconstructed, and re-verified until the semantic mismatch is resolved. Once alignment is achieved, the error flag switches to *No*, signaling successful correction.

### A Case Study of Iterative Correction

**Iteration 0. Initialize the code of the constraint:**

```
{
    "description": "Each pattern j
        ↪ should generate rolls with
        ↪ widths to fit within the
        ↪ RollWidth",
    "code": "for j in range(NumPatterns
        ↪ ):\n model.addConstr(sum(
        ↪ NumRollsWidth[i][j] * Widths[
        ↪ i] for i in range(NumWidths))
        ↪  <= RollWidth * NumRollsCut[j
        ↪ ])",
    "error": ""
},
```

**Iteration 1. Generate new description and verify:**

```
{
    "description_new": "For each
        ↪ pattern j, the sum of rolls
        ↪ produced must be arranged so
        ↪ that their total width does
        ↪ not exceed the width of the
```

```
        ↪ raw roll times the number of
        ↪ rolls cut using that pattern.
        ↪ "
    "error": "YES",
},
```

**Iteration 2. Update code and repeat the verify:**

```
{
    "code": "for j in range(NumPatterns
        ↪ ):\n model.addConstr(sum(
        ↪ NumRollsWidth[j][i] * Widths[
        ↪ i] for i in range(NumWidths))
        ↪  <= RollWidth)",
    "description_new": "Each pattern j
        ↪ must operate within the
        ↪ confines of RollWidth,
        ↪ dictating that the summarized
        ↪  width obtained from the
        ↪ rolls in that pattern remains
        ↪  within the roll's total
        ↪ width constraint."
    "error": "NO",
},
```

## 5. Conclusion

We presented SAC-Opt, a backward semantic-anchored correction framework for optimization modeling that explicitly addresses semantic inconsistencies in LLM-based models. By aligning reconstructed anchors from generated models with the original task description, SAC-Opt itera-

tively corrects only the mismatched components, driving convergence toward semantically faithful solutions. This backward, anchor-guided refinement moves beyond solver-driven syntactic checks, enabling fine-grained correction of constraints and objective without additional training or supervision. Experiments on seven public datasets demonstrate an average accuracy gain of 7.7%. These findings underscore the reliability of our semantic-anchored correction framework for LLM-based optimization workflows.

## Acknowledgements

This work is supported by the Early Career Scheme (No.CityU 21219323) and the General Research Fund (No.CityU 11220324) of the University Grants Committee (UGC), the NSFC Young Scientists Fund (No.9240127), and the Donation for Research Projects (No.9229164 and No.9229216). This work was done while the first author was an intern at Huawei Noah's Ark Lab.

## Impact Statement

This paper presents work whose goal is to advance the field of optimization modeling. There are many potential societal consequences of our work, none which we feel must be specifically highlighted here.

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

# A. Appendix

## A.1. Example of Structured Data

> ### An Example of Extracted Structured Data
>
> **The problem description:** This is a cutting stock problem. Given a roll of width 'RollWidth' and a set of widths 'Width' to be cut. Each width 'i' has a certain number of Orders 'Orders_{i}'. There are 'NumPatterns' patterns and each pattern 'j' has a certain number of rolls of each width 'i' 'NumRollsWidth_{i, j}'. The problem aims to minimize the total number of raw rolls cut. It is constrained that for each width 'i', the total number of rolls cut meets the total Orders. How to decide the number of rolls cut using each pattern 'j'?
>
> - - - - - - - - - - - - - - - - - - - - - - - - - - - - - - - - - - - - - - - - - - - - - - - - - - - - - - - -
>
> **The Structured data:**
>
> ```
> {
>     "parameters": [
>         {
>             "definition": "The width of the raw roll to be cut",
>             "symbol": "RollWidth",
>             "value": "",
>             "shape": [],
>             "code": "RollWidth = data[\"RollWidth\"] # scalar parameter"
>         },
>         {
>             "definition": "The set of widths to be cut",
>             "symbol": "Widths",
>             "value": "",
>             "shape": [
>                 "NumWidths"
>             ],
>             "code": "Widths = np.array(data[\"Widths\"]) # ['NumWidths']"
>         },
>         {
>             "definition": "The number of orders for each width",
>             "symbol": "Orders",
>             "value": "",
>             "shape": [
>                 "NumWidths"
>             ],
>             "code": "Orders = np.array(data[\"Orders\"]) # ['NumWidths']"
>         },
>         {
>             "definition": "The number of cutting patterns",
>             "symbol": "NumPatterns",
>             "value": "",
>             "shape": [],
>             "code": "NumPatterns = data[\"NumPatterns\"] # scalar parameter"
>         },
>         {
>             "definition": "The number of rolls of each width used in each pattern",
>             "symbol": "NumRollsWidth",
>             "value": "",
>             "shape": [
>                 "NumPatterns",
>                 "NumWidths"
>             ],
>             "code": "NumRollsWidth = np.array(data[\"NumRollsWidth\"]) # ['
>                 ↪ NumPatterns', 'NumWidths']"
>         },
>         {
>             "definition": "The number of different widths available to be cut",
>             "symbol": "NumWidths",
> ```

```
48              "value": "",
49              "shape": [],
50              "code": "NumWidths = data[\"NumWidths\"] # scalar parameter"
51          }
52      ],
53      "constraints": [
54          {
55              "description": "For each width i, the total number of rolls cut using all
                   ↪ patterns must meet or exceed the total number of Orders for that
                   ↪ width",
56              "code": null,
57              "error": ""
58          },
59          {
60              "description": "Each pattern j should generate rolls with widths to fit
                   ↪ within the RollWidth",
61              "code": null,
62              "error": ""
63          },
64          {
65              "description": "Number of raw rolls cut using each pattern j (NumRollsCut)
                   ↪ must be non-negative",
66              "code": null,
67              "error": ""
68          }
69      ],
70      "variables": {
71          "NumRollsCut": {
72              "shape": [
73                  "NumPatterns"
74              ],
75              "type": "integer",
76              "definition": "The number of raw rolls cut using each pattern"
77          }
78      },
79      "objective": {
80          "description": "\"The goal is to minimize the total number of raw rolls cut
                   ↪ \"",
81          "code": null,
82          "error": ""
83      },
84 }
```

**The data.json file associated with the parameters:**

```
1  {
2      "RollWidth": 10,
3      "Widths": [
4          2,
5          3,
6          5
7      ],
8      "Orders": [
9          4,
10         2,
11         2
12     ],
13     "NumPatterns": 2,
14     "NumRollsWidth": [
15         [
16             1,
```

```
17              2,
18              0
19          ],
20          [
21              0,
22              0,
23              1
24          ]
25      ],
26      "NumWidths": 3
27  }
```

## A.2. The Prompt of Constraint Reconstruction

```
1  prompt_constraints_language = """
2  You are an expert in optimization modeling. Here is the natural language description of
      ↪ an optimization problem:
3
4  {description}
5
6  You are given a constraint implemented in {solver} code and an example natural language
      ↪ description that serves only as a reference for sentence structure and length.
      ↪ Your task is to generate a **new** natural language description that:
7
8
9  1. **Is derived strictly from the given code** - do not assume information not present
      ↪ in the code.
10 2. **Maintains the structure, length, and complexity of the example description**, but
      ↪ is reworded.
11 3. **Does not directly copy the example text** - use a natural rephrasing while
      ↪ preserving accuracy.
12
13 The example description for the constraint is (For Structure & Length Reference Only,
      ↪ NOT for Content Copying):
14
15 -----
16 {constraint}
17 -----
18
19 Here is the code for the constraint:
20
21 -----
22 {constraint_code}
23 -----
24
25 Here is a list of parameters that are related to the constraint:
26
27 -----
28 {params}
29 -----
30
31 Here is a list of variables related to the constraint:
32
33 -----
34 {vars}
35 -----
36
37 The new description should be written in the following format:
38
39 CONSTRAINT:
40 =====
```

```
41  new natural language description for translating the constraint. (The description should
        ↪  be fully based on the code and should match the structure and length of the
        ↪ example description.)
42  =====
43
44  - Do not generate anything after the last =====.
45  - Do not include any additional information or explanations.
46
47  First reason about how the natural language description should be written, and then
        ↪ generate the output.
48
49  Please take a deep breath and think step by step. You will be awarded a million dollars
        ↪ if you get this right.
50
51  """
```

### A.3. The Prompt of LLM-based Verification

```
1   prompt_constraints_language_coverage = """
2   You are an expert in optimization modeling.
3
4   You task is to judge the consistency of the new generated description and the original
        ↪ description of the same constraint.
5
6   The original description is:
7   -----
8   {constraint}
9   -----
10
11  The new description is:
12  -----
13  {constraint_new}
14  -----
15
16  Please respond with "YES" if the two descriptions are consistent, and "NO" if they are
        ↪ not.
17
18  The asnwer should be in the following format:
19
20  ANSWER:
21  =====
22  YES or NO (ONLY one word and the answer should be in capital letters)
23  =====
24
25  - Do not generate anything after the last =====.
26  - Do not include any additional information or explanations.
27
28  Please take a deep breath and think step by step. You will be awarded a million dollars
        ↪ if you get this right.
29
30  """
```

### A.4. Discussion of Convergence

**Case Study on Convergence.**  To illustrate the convergence behavior of SAC-Opt, we present a representative example from the *flowshop scheduling* problem in the ComplexOR dataset. Following Sec. 3.5, we treat each misaligned constraints as an element of the *error set*. At the initial iteration, the total 8 constraints are treated as 8 initial errors. As shown in Figure 2, the cardinality of the error set decreases steadily with each iteration, eventually reaching 0. This demonstrates that SAC-Opt progressively eliminates inconsistencies between the generated code and the problem semantics, ultimately achieving convergence.

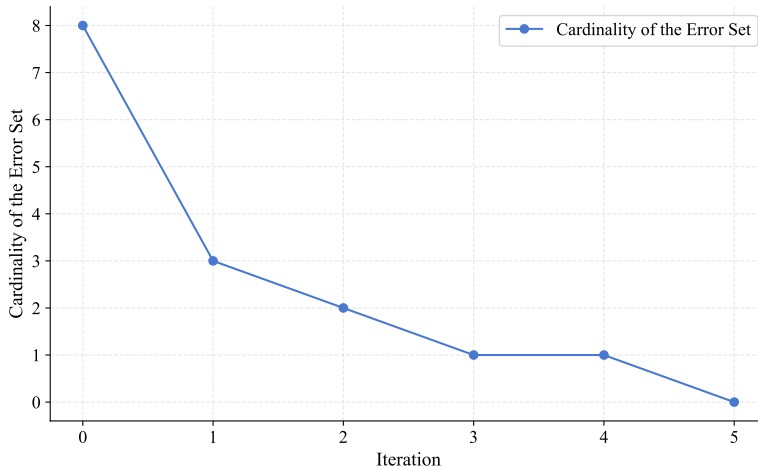

*Figure 2.* Comparison of cardinality of the error set and iteration count in SAC-Opt. Here the cardinality of the error set refers to the number of misaligned semantic anchors in the error set.

*Table 5.* Comparison of accuracy and average run time (in seconds) between SAC-Opt and the best baseline. The Impr. row shows relative accuracy gains, and the Diff. row reports runtime differences with respect to the baseline, where ↑ indicates an increase and ↓ a decrease.

| Metric | Method | NL4OPT | IndustryOR | EasyLP | ComplexLP | NLP4LP | ReSocratic | ComplexOR |
|---|---|---|---|---|---|---|---|---|
| | Best-baseline | 79.8% | 54.0% | 92.4% | 52.1% | 89.8% | 81.0% | 52.2% |
| Accuracy | SAC-Opt | 86.8% | 63.7% | 96.5% | 79.6% | 94.0% | 88.7% | 58.9% |
| | Impr. | 7.0 ↑ | 9.7 ↑ | 2.1 ↑ | 21.9 ↑ | 4.2 ↑ | 7.7 ↑ | 1.8 ↑ |
| | Best-baseline | 64.67 | 113.08 | 88.79 | 8.22 | 71.86 | 74.23 | 68.68 |
| Run Time (s) | SAC-Opt | 78.43 | 79.00 | 92.88 | 40.96 | 73.97 | 79.85 | 42.02 |
| | Diff. | 13.76 ↑ | 34.08 ↓ | 4.09 ↑ | 32.74 ↑ | 2.11 ↑ | 5.62 ↑ | 26.66 ↓ |

**Efficiency Analysis.** Beyond convergence in individual cases, we also assess the efficiency and generality of SAC-Opt across both easy and hard datasets. Detailed timing comparisons can be found in Appendix A.8. Table 5 merges results from Tables 1 and 7, comparing accuracy and average run time between SAC-Opt and the best baseline. Problem difficulty naturally affects convergence speed since easier tasks settle faster while harder ones require longer refinement, so we report averaged run time for fairness. The results show that SAC-Opt consistently improves modeling accuracy across all datasets, with particularly large gains on the more complex tasks (e.g., IndustryOR and ComplexLP). In terms of efficiency, the overhead remains modest, and in some datasets SAC-Opt even reduces total run time compared with the baseline. These findings confirm that SAC-Opt is both effective and efficient, delivering robust convergence and substantial improvements even on challenging real-world optimization problems.

## A.5. The Statistics of The Datasets

The dataset statistics are summarized in Table 6.

*Table 6.* The statistics of the datasets. The unit for description length is characters, and we report both the mean and standard deviation.

| Dataset | Description Length | # Instances | Multi-dimensional Parameters | Type |
|---|---|---|---|---|
| NL4OPT | 532.4 ± 103.0 | 214 | ✗ | Easy |
| IndustryOR | 554.7 ± 395.2 | 42 | ✓ | Hard |
| EasyLP | 1041.4 ± 257.7 | 545 | ✗ | Easy |
| ComplexLP | 504.7 ± 276.3 | 111 | ✓ | Hard |
| NLP4LP | 532.9 ± 108.1 | 178 | ✓ | Easy |
| ReSocratic | 554.2 ± 217.6 | 403 | ✓ | Hard |
| ComplexOR | 660.8 ± 197.2 | 18 | ✓ | Hard |

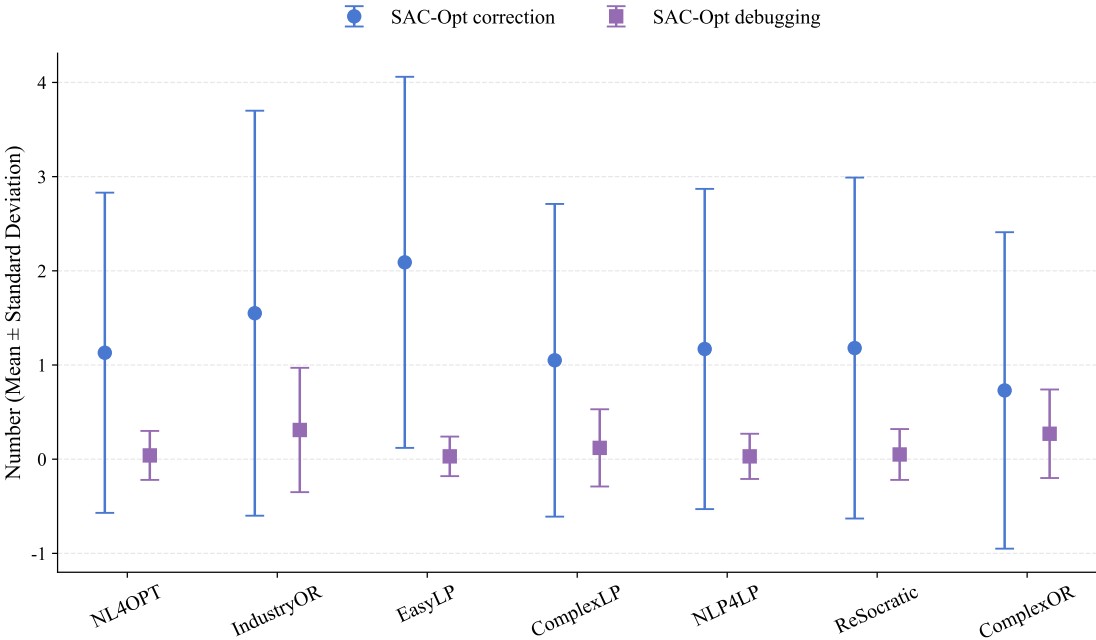

*Figure 3.* Comparison of average correction and debugging numbers in SAC-Opt.

## A.6. Discussion of Structured Data Extraction

SAC-Opt depends on the accuracy of the structured data extraction, which serves as the foundation for all downstream semantic reasoning. We acknowledge that semantic anchor extraction is an important and non-trivial task, yet it is not the central focus of this paper. Our contribution is to address the gap left by prior solver-driven approaches by proposing SAC-Opt, a backward-guided correction framework that grounds optimization modeling in problem semantics. In other words, SAC-Opt is not designed to solve the extraction task itself, but rather to preserve semantic fidelity even when extraction is imperfect, thereby ensuring that the resulting models remain aligned with the original problem intent.

To guarantee input quality and fairness in evaluation, we adopt the state-of-the-art extraction strategy from OptiMUS-0.3 (AhmadiTeshnizi et al., 2024a), which employs reflective prompting and confidence-based feedback to enhance the reliability and quality of the structured data. Importantly, the same extraction pipeline is used for all methods evaluated in this study, ensuring a consistent setting that isolates the correction capability of SAC-Opt. Experimental results further show that extraction noise is not the main limiting factor: on relatively simple datasets such as NL4OPT, EasyLP, NLP4LP, and ReSocratic, the average accuracy reaches 91.5%, confirming that structured data extraction is already highly reliable in practice.

To better assess the potential impact of extraction errors, we manually reviewed three challenging datasets and observed high accuracy in the structured data extraction stage, averaging above 94%: IndustryOR (4 errors out of 42), ComplexLP (3 out of 111), and ComplexOR (1 out of 18). Most issues involved minor misidentification of parameters or variables, while constraints and objective, the critical semantic anchors, were almost always extracted correctly. These findings provide strong evidence that SAC-Opt remains robust in practice and that its backward semantic correction delivers significant value beyond the extraction stage.

## A.7. Analysis of Correction and Debugging Numbers

To gain deeper insight into the behavioral differences between SAC-Opt's semantic correction and code-level debugging modules, we compare their average numbers across all datasets. As shown in Figure 3, the average number of semantic correction per instance is approximately 1.27, while debugging is invoked far less frequently, with an average of only 0.12. This significant gap emphasizes the dominant role of semantic correction in aligning model behavior with the intended task semantics. Unlike debugging, which passively reacts to execution failures, correction actively enforces semantic fidelity during the modeling process.

*Table 7.* Average run time (in seconds) comparisons of different methods.

| Method | NL4OPT | IndustryOR | EasyLP | ComplexLP | NLP4LP | ReSocratic | ComplexOR |
|---|---|---|---|---|---|---|---|
| Standard | 5.30 | 8.64 | 5.62 | 8.22 | 6.00 | 6.30 | 6.77 |
| CoT | 7.55 | 9.00 | 7.69 | 10.16 | 8.00 | 8.65 | 9.25 |
| CoE | 69.68 | 78.31 | 88.79 | 70.97 | 60.26 | 80.45 | 68.68 |
| CAFA | 7.52 | 9.94 | 7.56 | 9.48 | 8.66 | 8.11 | 9.22 |
| Reflexion | 8.32 | 14.26 | 9.34 | 14.28 | 9.28 | 9.28 | 11.64 |
| OptiMUS-0.2 | 59.41 | 55.20 | 59.41 | 48.63 | 62.87 | 51.05 | 84.63 |
| OptiMUS-0.3 | 64.67 | 113.08 | 82.60 | 89.61 | 71.86 | 74.23 | 52.96 |
| SAC-Opt-LLM | 78.43 | 80.20 | 92.88 | 40.96 | 73.97 | 79.85 | 42.02 |
| SAC-Opt-Sim | 198.82 | 209.68 | 183.89 | 173.76 | 208.67 | 174.99 | 66.58 |

## A.8. Run Time Comparison

Table 7 reports the average run time of each method across seven datasets. We have the following observations. First, simple inference methods such as Standard, CoT, and CAFA are highly efficient, with average run time around 6 to 7 seconds per instance. Their low computational overhead makes them suitable for fast but shallow modeling scenarios. Second, complex frameworks such as CoE, OptiMUS, and SAC-Opt require significantly more time due to iterative reasoning and correction. SAC-Opt consistently achieves the highest modeling accuracy, but its run time is less favorable on simpler datasets like EasyLP and NLP4LP, where semantic verification may be unnecessary when the initial generation is already correct. Third, LLM-based verification outperforms similarity-based verification in both accuracy and overall run time, but incurs a higher cost per call. In contrast, similarity-based methods are cheaper per problem but slower in total due to repeated correction operations. Future work may explore strategies to better balance verification quality with computational efficiency under different deployment constraints.

## A.9. Extended Statistical Analysis of Accuracy

*Table 8.* Accuracy (mean ± standard deviation, with % omitted) comparisons of different methods over five runs under a unified evaluation setting.

| Method | NL4OPT | IndustryOR | EasyLP | ComplexLP | NLP4LP | ReSocratic | ComplexOR |
|---|---|---|---|---|---|---|---|
| Reflexion | 68.2 ± 1.7 | 49.0 ± 3.9 | 85.8 ± 1.7 | 43.2 ± 2.8 | 82.4 ± 2.1 | 76.1 ± 1.1 | 42.2 ± 4.4 |
| OptiMUS-0.2 | 69.2 ± 1.9 | 43.8 ± 5.8 | 89.2 ± 1.1 | 45.8 ± 2.8 | 86.5 ± 1.8 | 75.8 ± 1.3 | 48.9 ± 4.2 |
| OptiMUS-0.3 | 79.8 ± 2.1 | 54.3 ± 2.8 | 92.4 ± 1.5 | 52.1 ± 2.3 | 89.8 ± 2.0 | 81.0 ± 1.9 | 52.2 ± 5.7 |
| SAC-Opt | **86.8 ± 1.5** | **63.8 ± 3.2** | **96.5 ± 0.5** | **79.6 ± 2.5** | **94.0 ± 1.5** | **88.7 ± 1.7** | **58.9 ± 5.7** |

Table 8 reports the mean and standard deviation over five independent runs for some methods under a unified setting that uses the same extraction pipeline, the GPT-4o backbone, and an identical accuracy metric. Across all seven datasets, SAC-Opt achieves the highest average accuracy while maintaining variance that is comparable to or lower than the baselines. Reflexion, OptiMUS-0.2, and OptiMUS-0.3 show larger fluctuations on datasets such as IndustryOR and ComplexOR, indicating less stable performance across seeds. In contrast, SAC-Opt delivers strong and consistent results across runs, reinforcing that the improvements reported in the main paper are robust and not due to randomness or evaluation inconsistencies.

