# OpenReview forum: "SAC-Opt: Semantic Anchors for Iterative Correction in Optimization Modeling"
_ICML.cc/2026/Conference — ICML 2026 regular_

### Official Review · Reviewer_XYnZ · 2026-03-06

**Soundness:** 2
**Presentation:** 3
**Significance:** 2
**Originality:** 3
**Overall Recommendation:** 4
**Confidence:** 3

**Summary:**

The paper introduces SAC-Opt, a backward-guided correction framework designed to eliminate "silent" semantic errors in LLM-generated optimization models. While traditional approaches rely on solver-driven feedback—which often fails to catch logical flaws in syntactically correct code—SAC-Opt grounds the correction process in the original problem semantics. The framework functions by aligning "semantic anchors" extracted from the natural language description with those reconstructed from the generated code, selectively refining only the mismatched components to ensure logical fidelity.

This anchor-driven design enables fine-grained adjustment of constraint and objective logic without requiring additional training or supervision. In evaluations across seven public datasets, SAC-Opt demonstrated a 7.7% average improvement in modeling accuracy, with performance gains reaching as high as 21.9% on the ComplexLP dataset.

**Compliance With Llm Reviewing Policy:**

Affirmed.

**Final Justification:**

The rebuttal addressed my main concerns. I still think empirical comparison with baselines would significantly strengthen the paper, but I am OK with the explanation in the rebuttal.

**Key Questions For Authors:**

1. Could you comment on the comparison to fine tuning?

2. In your experiments, how many instances failed to reach consistency before the iteration limit? For these cases, what is the impact on the final output? Additionally, how did you determine the optimal iteration limit to balance efficiency and accuracy?

3. Could you provide the API overhead of the SAC-Opt and compare it with the baseline?

**Limitations:**

No, the authors should explicitly address the relationship between semantic anchor alignment and logical correctness. Furthermore, the paper would benefit from a dedicated discussion on the trade-offs between improved accuracy and the increased computational overhead inherent in the iterative process.

**Strengths And Weaknesses:**

**Strengths**

- Good Experiment Results: Experimental results consistently demonstrate that the proposed method outperforms all established baselines across every tested dataset.

**Weaknesses**
- Unjustified correction loop design. The main innovation in the paper is to extract semi-structured text from both the natural language text description and the program, and compare their consistency to produce feedback to program generation. However, it is unclear why this feedback loop is the optimal design choice. If the LLM can precisely extract semantics anchors, why would it fail to generate correct programs? Is the accuracy improvement simply a fact of more token consumption? Would fine tuning the model on the task of generating programs from the semantics anchors yield a similar or even larger improvement?

- Lack Analysis of Convergence and Iteration Limits: The methodology relies on an iterative process of code generation until semantic anchors align or a maximum iteration limit is reached. The paper lacks data on how many cases failed to achieve consistency within this limit. Furthermore, it remains unclear how the choice of iteration limit affects the trade-off between model accuracy and computational efficiency.

- Potential Unfair Baseline Comparison: The framework requires multiple LLM generation cycles, raising significant concerns regarding API overhead and computational cost. More importantly, the comparison with baselines appears potentially unfair; since SAC-Opt benefits from multiple iterations, comparing it against single-pass or lower-budget baselines makes the experimental results less persuasive. Without a "cost-equivalent" baseline, it is difficult to discern whether the improvements stem from the anchor mechanism or simply from the increased number of attempts. The same applies to the ablation study.

---

> ### Author Rebuttal · Authors · 2026-03-30
>
> Thank you for the thoughtful questions. We agree that clearer iteration statistics and a more explicit API-budget analysis would strengthen the presentation. At the same time, we respectfully think the current paper already supports its main claims through an explicit anchor-level correction algorithm, controlled comparisons, ablations, cross-model generalization, and run time analysis.
>
> **W1&Q1:** We respectfully clarify that the correction loop is motivated by a concrete failure mode. Solver-driven pipelines can produce code that is executable yet semantically wrong, because solver feedback checks syntax and feasibility, but not whether the encoded constraints and objective match the original intent. SAC-Opt therefore does not simply retry generation. It reconstructs semantic anchors from code, identifies the misaligned error set, and selectively regenerates only those components. We do not claim this is the globally optimal loop among all possible designs. Our claim is narrower: this design directly targets silent semantic errors, and the controlled results show that it improves end-to-end correctness.
>
> This also explains why accurate anchor extraction does not guarantee correct solver code. Extraction and solver-specific code generation are different tasks. Appendix A.6 shows that structured extraction is already highly reliable, with above 94% accuracy on three challenging datasets, while downstream semantic gaps still remain. We also respectfully disagree that the improvement is simply a fact of "more token consumption." If extra attempts were the main reason, one would not expect better performance with fewer correction calls and lower run time, which is exactly what Table 4 shows for LLM-based verification. Table 2 also shows that removing semantic correction causes a much larger drop than removing debugging. Appendix A.7 further reports that the average number of semantic correction calls is only about 1.27, far below $T_{max}=5$. On fine-tuning, we respectfully view it as complementary rather than a direct substitute. Fine-tuning from semantic anchors to solver code would require paired supervision and training, while SAC-Opt is a training-free inference framework.
>
> **W2&Q2:** We agree that corpus-level iteration statistics would be informative, but we respectfully disagree that the paper lacks convergence structure. The algorithm has an explicit stopping rule: it terminates when the error set is empty or when $T_{max}$ is reached. Appendix A.4 shows a representative case in which the error set decreases to zero. Appendix A.7 also reports an average of only 1.27 semantic correction calls per instance, far below $T_{max}=5$.
>
> We also want to be precise about what is and is not reported. The current submission does not separately report how many instances hit $T_{max}$, and it does not include a dedicated sweep over $T_{max}$. We agree both would be informative. Still, cases that remain partially inconsistent at the cap do not inflate performance, because an instance is counted as correct only if the generated code executes successfully and returns the correct objective value and the correct optimal solution. We also do not claim that $T_{max}=5$ is globally optimal. It is used as a practical upper bound, and the observed averages suggest that typical behavior remains well below it.
>
>
> **W3&Q3:** We appreciate the concern about cost and fairness. However, we respectfully disagree that the current baseline comparison is fundamentally unfair. To ensure a rigorous and fair comparison, we adopt a unified evaluation protocol: all methods use the same GPT-4o backbone, the same cleaned datasets, the same shared extraction pipeline, and the same cap of three debugging attempts where applicable. The same logic also applies to the ablation study, which isolates the removal of semantic correction within the same framework rather than comparing unrelated budgets.
>
> The current submission reports run time rather than token or API-call counts, and we use run time as the system-level proxy for overhead. This overhead is real. SAC-Opt is more expensive than very cheap single-pass methods. At the same time, it is not simply winning by spending more compute than all stronger baselines. For example, on IndustryOR and ComplexLP, SAC-Opt takes 80.20s and 40.96s, while OptiMUS-0.3 takes 113.08s and 89.61s. Table 4 also shows that better semantic verification improves both accuracy and efficiency. A cost-equivalent retry baseline would certainly be informative, but its absence is different from showing that the current gains come only from repeated attempts.
>
> Thank you again for the constructive feedback. We hope these clarifications make the motivation, convergence behavior, and cost trade-offs of SAC-Opt more transparent.

---

> > ### Author Rebuttal · Reviewer_XYnZ · 2026-04-02
> >
> > Thank you for the clarification. While I still think emprical comparison with baselines would significantly strengthen the paper, the current explanation is acceptable.

---

> > > ### Author Response · Authors · 2026-04-02
> > >
> > > Thank you very much for raising your score and for indicating that our clarification adequately addressed your concerns. We sincerely appreciate your constructive suggestion that stronger empirical comparisons with baselines would further strengthen the paper, and we will keep this valuable point in mind as we improve the final version. Thank you again for your thoughtful review.

---

### Official Review · Reviewer_vm8j · 2026-03-10

**Soundness:** 3
**Presentation:** 2
**Significance:** 1
**Originality:** 3
**Overall Recommendation:** 4
**Confidence:** 4

**Summary:**

The paper is concerned with repair of mathematical optimization code using an iterative LLM-based procedure, which improves modeling accuracy on a range of benchmarks.

**Compliance With Llm Reviewing Policy:**

Affirmed.

**Final Justification:**

Thank you to the authors for the thoughtful rebuttal. After reading their replies also to the other reviwers I have increased some scores and overall assessment.
I still think that some ambitious claims could be toned down (perhaps my NLP background shows in that I'm particularly allergic to a too-liberal use of the term "semantic").

Edit 2: If the latest rebuttal remarks on the scope of the evaluation protocol makes it to the final manuscript, the bulk of my concerns wrt the soundness of this work will have been put to rest.

**Key Questions For Authors:**

No urgent questions beyond what I wrote above.

**Limitations:**

There is no Limitations discussion, but no foreseeable negative societal impact either.

**Strengths And Weaknesses:**

Soundness: There is an experimental evaluation of an iterative algorithm, and an accuracy metric goes down, but it's hard to assess some claims, as these are shrouded in fuzzy terms such as "semantic verification", "semantic alignment". The closest we get to a formalization of "semantic alignment" is via LLM based verification and with a pretrained vector embedding model (Sec. 3.5), but it is not shown whether these two methods are calibrated with each other.  Much of the method effectiveness hinges on LLM prompts, therefore some prompt ablations would be needed but are not given. To do justice to the central claim of the paper, that this method "detects silent semantic errors that solver-driven checks cannot capture" (L 110) would require natural data on these modeling failure modes (perhaps obtained by pair programming with a group of OR graduate students) and a more detailed experimental evaluation.

Presentation: The amount of notation introduced does not justify its use; in particular, writing a LLM string transformation pass as a deterministic function f is not rigorous.  Too much space is devoted to "pitching" the method rather than discussing it: Section 3.2 last paragraph, the last 2 paragraphs of Sec. 3.3 and the last of Sec. 3.4 do not add weight to the paper and should be removed in my opinion.

Significance: The paper is very tangentially related to ML, via its use of an LLM in the repair loop. The proposed method is not learning-based, and does not benefit from data (in its present form at least). Its significance would be greatly increased by using natural data and representing "intent" in a more formal way.

Originality: It seems this work follows incrementally from previous work, and (again) the novelty should lie in an ill-defined semantic alignment.

---

> ### Author Rebuttal · Authors · 2026-03-30
>
> Thank you for the detailed comments. We agree that the paper would be stronger with additional analyses such as prompt sensitivity. However, we respectfully disagree that these useful extensions should be elevated into objections to soundness, significance, or originality. The current submission already provides a formal method, a reproducible evaluation protocol, and empirical evidence for the core claim.
>
> **On Soundness**
> "Semantic alignment" and "semantic verification" are not vague intuitions in the paper. Fig. 1 and Sec. 3.4 explicitly define the semantic anchors as the constraints and objective, and Secs. 3.4-3.5 formalize the loop through structured intent (Eq. 2), reconstruction from code (Eq. 7), binary consistency verification (Eq. 8), the error set (Eq. 11), selective regeneration, and stopping at $T_{max}$ in Algorithm 1. Thus, the formalization is not limited to the verifier in Sec. 3.5; it is the full anchor-level reconstruction-verification-correction procedure.
>
> The concern about whether the LLM-based verifier and the pretrained-vector similarity verifier are "calibrated" misstates their role. They are not fused scores that must be jointly calibrated, but two alternative implementations of the same semantic consistency function. The relevant issue is therefore empirical comparison, already reported in Sec. 4.7 / Table 4 through accuracy, run time, correction counts, and debugging attempts.
>
> We agree that prompt ablations would be useful. At the same time, the paper already discloses the reconstruction and verification prompts in Appendices A.2-A.3, and the core contribution is the backward correction architecture rather than a single prompt. This is also consistent with the experiments: removing semantic correction causes a much larger drop than removing debugging (Table 2), and the gains persist for both GPT-4o and Qwen2.5-72B-Instruct (Table 3). The evaluation metric already captures the "silent semantic errors" at issue: an output is counted as correct only if the code executes and returns both the correct optimal objective value and the correct optimal solution, so solver-executable but semantically wrong programs are already counted as failures.
>
> We agree that natural data on these failure modes would be valuable future work. However, we do not believe they are prerequisites for the current claim. All methods are evaluated on seven cleaned public benchmarks under a shared extraction pipeline and a fixed debugging budget, providing a reproducible and fair comparison setting. The appendix further supports the loop behavior: Appendix A.4 gives a convergence case; Appendix A.6 shows extraction accuracy above 94% on three challenging datasets, with constraints/objective almost always correct; and Appendix A.7 shows that semantic correction is the dominant mechanism in practice, with about 1.27 correction calls versus 0.12 debugging calls on average.
>
> **On Presentation**
> We agree that some motivational paragraphs can be shortened. However, the paper does not model an LLM semantic pass as a deterministic function. Sec. 3.3 explicitly distinguishes the deterministic template-rendering function $f_{trans}^{det}$, used only for parameters and variables, from the agent-based semantic translation function $f_{trans}^{agent}$, used for constraints and objective. The notation is therefore used to specify module interfaces and pipeline composition, not to claim determinism of all LLM components.
>
> **On Significance**
> We respectfully disagree that the work is only tangentially related to ML. The paper is explicitly positioned in the inference-framework line of LLM-based optimization modeling and formalizes problem intent as S=(P,V,C,O) to enable modular verification and correction. The method is training-free rather than learning-based, but that is the intended setting. Empirically, SAC-Opt achieves the best overall results on seven public benchmarks, with a 7.7% average gain and a 21.9% gain on ComplexLP; these gains also transfer from GPT-4o to Qwen2.5-72B-Instruct, indicating that the improvement comes from the correction mechanism rather than one specific backbone. New natural datasets would certainly be valuable future work, but we do not believe their absence makes the current contribution insignificant.
>
> **On Originality**
> We respectfully do not believe the contribution is merely incremental, nor that the novelty rests only on an "ill-defined semantic alignment." The novelty is the backward reconstruction-verification-correction loop itself. Prior workflows mainly rely on forward generation, or solver-error-driven debugging. In contrast, SAC-Opt reconstructs semantic anchors from code, checks them against the original intent, and selectively regenerates only the misaligned components. This changes the source of correction signals from solver feedback to problem semantics. Consistently, under the same protocol, SAC-Opt outperforms strong solver-driven baselines such as Reflexion and OptiMUS-0.3.

---

> > ### Author Rebuttal · Reviewer_vm8j · 2026-04-02
> >
> > > The concern about whether the LLM-based verifier and the pretrained-vector similarity verifier are "calibrated" misstates their role. They are not fused scores that must be jointly calibrated, but two alternative implementations of the same semantic consistency function.
> >
> > Table 4 shows the two accuracy scores side by side. There is some correlation between the two, but a. it could be made quantitative with an R2 score or similar b. we do not know whether this trend corresponds to any human-correlated measure of relevance.

---

> > > ### Author Response · Authors · 2026-04-03
> > >
> > > Thank you again for this helpful follow-up and for your continued engagement with our work. We sincerely appreciate your constructive suggestions. We agree that Sec. 4.7 can be made clearer on this point. Below, we first quantify the relationship between the two verifier variants, and then clarify the scope of this evidence.
> > >
> > > First, we computed a quantitative summary from the seven dataset-level accuracy pairs in Table 4. The LLM-based and similarity-based variants are strongly correlated. The Pearson correlation is 0.962, with ($R^2 = 0.925$). The Spearman correlation is 0.929. In addition, the LLM-based verifier performs better on all seven datasets, with a mean absolute gain of 6.94 accuracy points. We will add this summary in the revision to make the trend in Table 4 explicit. At the same time, we want to be precise about the scope of this evidence. It provides dataset-level agreement evidence. It does not yet provide a human-annotated anchor-level calibration study.
> > >
> > > Second, our goal in Sec. 4.7 is to compare two practical automatic implementations of the same anchor-level consistency function ($\delta$). As defined in Sec. 3.5, $\delta_{\text{LLM}}$ and $\delta_{\text{sim}}$ are two alternative automatic choices inside the same correction loop. They are not two scores that are fused. For this reason, the relevant question here is their comparative behavior within the same pipeline, rather than joint calibration of fused scores. Appendix A.3 makes this concrete, since the LLM-based verifier is implemented as a strict YES or NO consistency judgment. We chose these two variants because they represent two practical automatic options within the same fully automated end-to-end pipeline. The similarity-based verifier is lightweight and easy to deploy. The LLM-based verifier provides a stronger automatic signal.
> > >
> > > We also agree that human-grounded validation would be useful additional evidence, and we will make this limitation clearer in the revision. At the same time, our framework is modular. Expert human assessment could in principle be incorporated into the same loop by replacing the automatic verifier module. Our current paper, however, focuses on the fully automated setting. In this setting, the external target is exact optimization correctness rather than a more open-ended notion of relevance. As defined in Sec. 4.3, a problem is counted as correct only if the generated code executes successfully and returns both the correct optimal objective value and the correct optimal solution.
> > >
> > > In this sense, the verifier is an internal guidance signal, while the final external validation is downstream task correctness. Table 4 therefore provides dataset-level agreement evidence, and the full benchmark results provide task-level external validation. Under the same protocol, the stronger automatic verifier is associated with better end-task correctness, even though neither verifier is claimed to be a perfect oracle for human semantic judgment.
> > >
> > > More generally, we view an oracle semantic consistency function as an upper bound for this family of correction loops. If verification were perfect, and if the correct anchor implementation were reachable within the search budget, the loop would recover a semantically aligned model. The comparison between $\delta_{\text{LLM}}$ and $\delta_{\text{sim}}$ should therefore be read as a comparison between two practical approximations to that upper bound.
> > >
> > > Thank you again for the careful reading and for pushing us to sharpen this part of the paper. We hope this clarifies the role of the two verifier variants and the scope of the evidence in the current paper.

---

### Official Review · Reviewer_KaVa · 2026-03-11

**Soundness:** 3
**Presentation:** 2
**Significance:** 2
**Originality:** 2
**Overall Recommendation:** 4
**Confidence:** 4

**Summary:**

This paper addresses the research question that how to mitigate semantic inconsistencies in LLM-based optimization modeling. They propose SAC-Opt with key idea that the framework extracts semantic anchors such as constraints and objectives, reconstructs them from generated code and compares them with the original intent to iteratively refine. Experiments on seven public datasets show consistent gains over prior baselines.

**Compliance With Llm Reviewing Policy:**

Affirmed.

**Final Justification:**

The rebuttal period addressed my main concerns about this paper.

**Key Questions For Authors:**

1. How reliable is the LLM-based semantic verifier, and does its performance correlate with true semantic correctness? To be more specific, did the authors observe cases of false positives where the verifier marked a reconstructed anchor as correct even though the underlying code was logically flawed?
2. Since all methods use a shared structured extraction pipeline, how much of the gain comes from SAC-Opt itself versus this preprocessing choice?
3. The paper mentions that convergence is achieved when all anchors are correctly represented. In practice, how many iterations does the model typically require to reach this state on datasets like IndustryOR or ComplexLP? Would it cause any latency issues?

**Limitations:**

yes

**Strengths And Weaknesses:**

Strenths:
1. The proposed refinement loop is easy to understand. By extracting semantic anchors and comparing them against reconstructed logic from the generated code, the method provides a structured way to detect semantic errors.
2. The empirical results demonstrate that SAC-Opt improves over prior baselines with large gains on ComplexLP. To be more specific, the average modeling accuracy improves by 7.7% across seven public datasets, with ComplexLP being 21.9%.

Weaknesses:
1. The method depends heavily on the quality of the verifier. Essentially it relies on the f_{agent}^{verif} function to determine semantic equivalence. The paper would benefit from a more detailed analysis of verifier reliability.
2. The comparison in the experiment part is a bit unclear. All methods use shared structured extraction, which is intended to help reducing preprocessing noise. However, as the framework is specifically built to leverage these extracted structured anchors for its iterative loop, this structured extraction may help SAC-Opt during evaluation.

---

> ### Author Rebuttal · Authors · 2026-03-30
>
> Thank you very much for your constructive and encouraging feedback. We sincerely appreciate your recognition that SAC-Opt provides a structured way to detect semantic errors and that the gains are especially meaningful on complex datasets such as ComplexLP. We also appreciate your thoughtful questions about verifier reliability, the role of shared structured extraction, and the practical behavior of the iterative loop. We address these points below.
>
> **W1&Q1: Verifier reliability and false positives.**
>
> We agree that verifier quality is important, since semantic consistency checking is central to SAC-Opt. At the same time, we do not assume that the verifier is a perfect oracle. The paper already studies this dependence at the system level in Sec. 4.7 / Table 4 by comparing LLM-based and similarity-based verification. The LLM-based verifier consistently achieves higher accuracy with fewer correction attempts and shorter run time. The similarity-based variant still outperforms most baselines on several challenging datasets. This suggests that the iterative correction architecture remains useful even under a lower-fidelity semantic signal. In this sense, the framework depends on verifier quality, but it is not brittle to verifier choice.
>
> Regarding false positives, we agree that this is an important concern, and we do not want to overclaim beyond what is explicitly reported. The current submission does not provide a separate per-anchor false-positive analysis. However, the verifier is used as an internal guidance signal rather than as the metric used to claim success. Our final evaluation is strict: an instance is counted as correct only if the generated code executes successfully and returns both the correct optimal objective value and the correct optimal solution. Therefore, even if the verifier occasionally marks a flawed anchor as consistent, that error would still be penalized by the main accuracy metric rather than artificially inflating the reported results.
>
> **W2&Q2: Shared structured extraction and whether it unfairly helps SAC-Opt.**
>
> We respectfully disagree that the shared structured extraction pipeline unfairly favors SAC-Opt. In fact, we adopt a shared pipeline precisely to ensure a rigorous and fair comparison by holding preprocessing fixed across methods. Otherwise, any improvement could be confounded with differences in extraction quality rather than reflecting the correction mechanism itself. This is why all methods operate on structured data produced by the same OptiMUS-0.3 extraction pipeline under a unified evaluation protocol.
>
> More importantly, the current paper already helps separate preprocessing effects from SAC-Opt's intrinsic gain. Appendix A.6 shows that extraction noise is not the main bottleneck in practice. Manual review on three challenging datasets gives above 94% extraction accuracy, and the critical semantic anchors, namely constraints and objective, are almost always extracted correctly. In addition, Table 2 provides a direct within-framework control. When semantic correction is removed while the same extraction remains fixed, performance drops substantially across all datasets. This strongly suggests that the observed gains come primarily from SAC-Opt's semantic correction loop rather than from the preprocessing choice.
>
>
> **Q3: Practical number of iterations and latency.**
>
> In practice, the reported correction counts suggest that the loop usually resolves misalignment well before $T_{max}=5$. Appendix A.7 reports an average of about 1.27 semantic correction calls per instance overall. Table 4 further shows that, with LLM-based verification, the average number of corrections is about 1.55 on IndustryOR and 1.05 on ComplexLP. This indicates that the typical behavior is around one to two semantic correction steps rather than long iterative chains.
>
> As for latency, Appendix A.8 and Table 7 already report the average run time for all methods. SAC-Opt-LLM takes 80.20s on IndustryOR and 40.96s on ComplexLP. This is naturally slower than simple single-pass methods. However, it remains competitive with strong iterative baselines, and in some cases it is faster, for example compared with OptiMUS-0.3 on both IndustryOR and ComplexLP. We therefore view the run time cost as a real but manageable trade-off for improved semantic fidelity, rather than evidence that the loop creates severe practical latency issues.
>
> Thank you again for your thoughtful and supportive review. We hope these clarifications make the role of the verifier, the rationale for the shared extraction setting, and the practical behavior of the correction loop more transparent.

---

> > ### Author Rebuttal · Reviewer_KaVa · 2026-04-02
> >
> > Thank you for the detailed response. I'll maintain my positive score 4.

---

> > > ### Author Response · Authors · 2026-04-03
> > >
> > > Thank you again for your thoughtful review and for the time and effort you have devoted to evaluating our work. We sincerely appreciate your positive assessment, and we would be very happy to clarify anything further if helpful.

---

### Official Review · Reviewer_iTfY · 2026-03-12

**Soundness:** 3
**Presentation:** 3
**Significance:** 2
**Originality:** 2
**Overall Recommendation:** 5
**Confidence:** 3

**Summary:**

The paper tries to make LLM-generated optimization code actually reflect what a problem means. Normal methods only fix things when the solver complains, which misses semantic issues. SAC-Opt fixes that by introducing semantic anchors for each constraint and objective. After generating code, it reconstructs the anchors from the code and compares them with the originals. Any mismatch gets iteratively corrected. Experiments on seven datasets show improvements, especially in complex cases. The authors present a notable theme of using semantic alignment rather than just solver feedback. Overall, the study's principal domain is applying LLMs for optimization modeling, and the backward correction seems like a clever way to catch logic mistakes before they mess up results.

**Compliance With Llm Reviewing Policy:**

Affirmed.

**Final Justification:**

I suggest that this paper be accepted. After carefully assessing the original manuscript's quality as well as the authors' reply, I think this work satisfies publishing requirements and shows adequate contributions that are expressed clearly.

**Key Questions For Authors:**

It might be interesting to see how SAC-Opt handles ambiguous or incomplete problem descriptions. Also, does the iterative loop ever get stuck if reconstruction keeps missing subtle logic?

**Limitations:**

yes

**Strengths And Weaknesses:**

**Strengths**:

- Focus on semantics rather than just syntax is unique. This is important because solver success only shows that the code is executable, not that it faithfully captures the problem intent. By focusing on semantic consistency between the text and the generated model, the paper addresses a deeper and more meaningful source of errors.
- Iterative, targeted corrections are more efficient than regenerating everything. Correcting only the mismatched constraints or objectives is more controlled than rewriting the entire optimization model from scratch. It also reduces the risk of breaking parts that were already generated correctly.
- Strong empirical results. The method is evaluated across seven datasets and shows especially clear gains on more complex problems, which supports the claim that semantic correction helps in difficult settings. The empirical improvements therefore feel both broad and practically meaningful.

**Weaknesses**:

- Extraction step could be fragile. Since the whole framework depends on the initial extraction of variables, constraints, and objectives, any mistake at that stage can propagate through the later correction loop. This means the final alignment may still be built on an imperfect representation of the original problem.
- Verification might be sensitive to unusual wording.

---

> ### Author Rebuttal · Authors · 2026-03-30
>
> Thank you very much for your positive and constructive feedback. We sincerely appreciate your recognition that semantic consistency is a deeper target than solver executability alone. We also appreciate your point that correcting only mismatched constraints or the objective is more controlled than full regeneration. We address your comments on extraction fragility, wording sensitivity, and edge cases below.
>
> **W1: Extraction step could be fragile.**
>
> We agree that structured data extraction is important, since it provides the semantic foundation for later stages. The goal of SAC-Opt is not to redesign the extraction stage, but to preserve semantic fidelity after extraction. For fairness, all methods use the same extraction pipeline from OptiMUS-0.3, so the effect of semantic correction is isolated from extraction quality. Appendix A.6 also suggests that extraction noise is not the main bottleneck in our experiments. On three challenging datasets, extraction accuracy is above 94%, and the critical semantic anchors, namely constraints and objective, are almost always extracted correctly. Most remaining issues are minor parameter or variable mismatches. Thus, extraction noise is a real limitation, but it does not appear to be the dominant source of error in our setting.
>
> **W2: Verification may be sensitive to unusual wording.**
>
> We agree that unusual wording can make semantic verification harder. This is a real limitation. Our design reduces this risk in two ways. First, SAC-Opt verifies each constraint or objective at the anchor level, instead of comparing whole programs. This makes the check more focused and less dependent on surface wording. Second, the LLM-based verifier is intended to judge semantic consistency rather than lexical overlap. Table 4 shows that it achieves higher accuracy with fewer correction steps and shorter run time than the similarity-based verifier. We will make this limitation clearer in the revision and note that more robust semantic verification for unusual wording is an important direction for future work.
>
> **Q1: Ambiguous or incomplete problem descriptions, and whether the loop can get stuck.**
>
> Thank you for this important question. In this work, we focus on public benchmark datasets that are relatively well-defined and interpretable. In these settings, the main challenge is to generate solver code that remains faithful to the stated intent. For fundamentally ambiguous or incomplete descriptions, there may be no single correct model even for a human. We agree that this is important, but it is beyond the main scope of the current paper. Our structured representation may still help by making parameters, variables, constraints, and objective explicit, which can surface ambiguity earlier in the pipeline.
>
> The loop also cannot run forever. SAC-Opt uses a strict maximum iteration limit of $T_{max}=5$. If some semantic mismatch remains, the correction phase stops and returns the latest code. In practice, repeated non convergence does not appear common. Appendix A.4 shows a representative case where the error set steadily decreases to zero. Appendix A.7 also shows that the average number of semantic correction steps is about 1.27 per instance. Final correctness is still evaluated strictly. The code must execute successfully and return both the correct optimal objective value and the correct optimal solution.
>
> We thank you again for your encouraging assessment and helpful suggestions. We hope these clarifications make the practical robustness and scope of SAC-Opt clearer.

---

> > ### Author Rebuttal · Reviewer_iTfY · 2026-04-02
> >
> > I have decided to maintain my original score, as I believe it already reflects a fair and sufficiently positive assessment of the work.

---

> > > ### Author Response · Authors · 2026-04-03
> > >
> > > Thank you again for your thoughtful review and for the time and effort you have devoted to evaluating our work. We sincerely appreciate your positive assessment, and we would be very happy to clarify anything further if helpful.

---

### Decision · Program_Chairs · 2026-04-30

**Decision:**

Accept (regular)

**Comment:**

This paper studies how to translate natural-language descriptions of an optimization problem into a formal mathematical model. The paper aims to address "silent modeling failures": models where the solver successfully executes, but the model does not match the ground-truth formulation of the natural language description. The key idea is to extract semantic anchors such as constraints and objectives, reconstruct them from the generated code, and compare them with the original intent to iteratively refine. While the initial round of reviews was mixed, the authors substantively engaged in the response period, and all reviewers increased their scores to the accept range. The authors should incorporate all promised changes in the revision.